# The Experimental and In Silico-Based Evaluation of NRF2 Modulators, Sulforaphane and Brusatol, on the Transcriptome of Immortalized Bovine Mammary Alveolar Cells

**DOI:** 10.3390/ijms25084264

**Published:** 2024-04-12

**Authors:** Hunter R. Ford, Massimo Bionaz

**Affiliations:** Department of Animal and Rangeland Science, Oregon State University, Corvallis, OR 97331, USA; hunter.ford@ttu.edu

**Keywords:** NRF2, bovine, sulforaphane, brusatol, transcriptome, MACT cells

## Abstract

Changes during the production cycle of dairy cattle can leave these animals susceptible to oxidative stress and reduced antioxidant health. In particular, the periparturient period, when dairy cows must rapidly adapt to the sudden metabolic demands of lactation, is a period when the production of damaging free radicals can overwhelm the natural antioxidant systems, potentially leading to tissue damage and reduced milk production. Central to the protection against free radical damage and antioxidant defense is the transcription factor NRF2, which activates an array of genes associated with antioxidant functions and cell survival. The objective of this study was to evaluate the effect that two natural NRF2 modulators, the NRF2 agonist sulforaphane (SFN) and the antagonist brusatol (BRU), have on the transcriptome of immortalized bovine mammary alveolar cells (MACT) using both the RT-qPCR of putative NRF2 target genes, as well as RNA sequencing approaches. The treatment of cells with SFN resulted in the activation of many putative NRF2 target genes and the upregulation of genes associated with pathways involved in cell survival, metabolism, and antioxidant function while suppressing the expression of genes related to cellular senescence and DNA repair. In contrast, the treatment of cells with BRU resulted in the upregulation of genes associated with inflammation, cellular stress, and apoptosis while suppressing the transcription of genes involved in various metabolic processes. The analysis also revealed several novel putative NRF2 target genes in bovine. In conclusion, these data indicate that the treatment of cells with SFN and BRU may be effective at modulating the NRF2 transcriptional network, but additional effects associated with cellular stress and metabolism may complicate the effectiveness of these compounds to improve antioxidant health in dairy cattle via nutrigenomic approaches.

## 1. Introduction

High-producing dairy cattle in the early postpartum period are susceptible to oxidative dysfunction, which is a condition that can lead to tissue damage and cause losses in milk production [1]. While free radicals that contribute to oxidative damage are constantly produced at low levels by normal cellular and immune processes, they are quickly converted into nonreactive products, like H_2_O, by different endogenous antioxidant systems like the glutathione-family enzymes, superoxide dismutase, and catalase [2]. During times of high metabolic or immune activity, the production of free radicals can overload antioxidant capacities and begin to damage cells via interactions with membranes, proteins, and DNA, leading to cell death [3].

For dairy cows, the transition from pregnancy to lactation is often accompanied by a shift into a negative energy balance and the mobilization of non-esterified fatty acids (**NEFAs**) as a response [4]. Using these NEFAs for energy can also lead to increases in reactive oxygen species (**ROS**) production by the mitochondria and, eventually, cell death, which in the mammary gland contributes to losses in milk production [5]. Oxidative dysfunction can also negatively impact the function of immune cells like macrophages and neutrophils, making animals more susceptible to other diseases, like mastitis [6].

Cells can effectively reduce the low levels of ROS that are normally produced during homeostatic metabolic and immune processes. This capability has been primarily linked to the activity of a wide array of antioxidant enzymes, many of which contain a common genetic sequence in their encoding DNA that is known as the antioxidant response element (**ARE**) [7] and is recognized mainly by the transcription factor nuclear factor erythroid 2-related factor 2 (**NRF2**) [8]. Much of the work regarding NRF2 has been conducted concerning its role in cancer, where it is often found to be constitutively activated, providing tumors with enhanced resistance to chemotherapeutics [9]. In nonruminant animals, NRF2 regulates the transcription of several genes involved in ROS elimination, glutathione synthesis and maintenance, NADPH generation, and cell survival [10], with the modulation of NRF2 being an effective mechanism for restoring redox homeostasis and preventing oxidative damage to cells and tissues [11].

Despite the importance of redox homeostasis for dairy cow production and health, very limited work has been performed to better understand the role of NRF2 in ruminant models. Promising work has shown that bovine cells respond to putative NRF2 agonists and that NRF2 activation can limit oxidative dysfunction and improve cell viability [12,13,14]. In monogastric models, the known targets of NRF2 are well-established [15]; however, very few studies have incorporated a direct measure of NRF2 activity or expanded their transcriptomic analysis of NRF2 modulation beyond these known target genes.

In ruminant animals, evidence indicates that NRF2 may be modulated via dietary components that may be supplemented in animals or found in pasture forages, including phenolic compounds, carotenoids, and terpenoids [16]. In a prior study, we demonstrated that the natural NRF2 agonist sulforaphane (**SFN**) and the natural NRF2 antagonist brusatol (**BRU**) are effective in modulating NRF2 in bovine mammary epithelial alveolar cells (**MACT**) [12]; thus, the nutrigenomic potential to improve the antioxidant response in ruminant animals via NRF2 exists. However, the consequences of NRF2 modulation by SFN and BRU in the bovine mammary gland have yet to be determined at the transcriptomic level. Such findings provide valuable insight into the efficacy of potential nutrigenomic approaches for improving antioxidant capacity in ruminant animals and the NRF2-controlled regulatory network.

To this aim, the objective of this study was to identify the genomic implications of NRF2 modulation in bovine cells by SFN and BRU using a whole-transcriptome approach. Given the broad similarities between the genetic sequences of NRF2 and its targets in ruminant and monogastric models, we expect that the effects of NRF2 modulation will be similar.

We hypothesize that NRF2 activation upregulates genes associated with antioxidant defense and cell survival, whereas the inhibition of NRF2 results in the reduced expression of important antioxidant genes, with perhaps an increase in the expression of genes associated with apoptosis and cell death.

## 2. Results

### 2.1. Differentially Expressed Genes Determined by RNA Sequencing

There were 16,888 transcripts that were detected with at least one raw read in at least one sample (Appendix A). When MACT cells were treated with SFN, the use of a false discovery rate (**FDR**) < 0.05 revealed 934 transcripts differentially expressed (**DEGs**), with 427 DEGs found to be upregulated and 507 DEGs downregulated compared to non-treated MACT cells (**CTR**). The treatment of MACT cells with BRU resulted in the identification of 3977 DEGs, with 2061 DEGs upregulated and 1916 DEGs downregulated. Comparing SFN to BRU treatment, 4312 transcripts were found to be differentially expressed, with 2093 DEGs upregulated and 2219 DEGs downregulated. Figure 1 shows the number of DEGs, including a fold-change cut-off. The complete results are available in Appendix A.

### 2.2. Expression of Putative NRF2 Target Genes

The top 30 up and downregulated DEGs in each comparison and the possible target genes of NRF2 are reported in Appendix A. The most upregulated protein-coding DEGs in SFN vs. CTR were *IL36* and *PHLDA2*, together with a large number of non-coding small nuclear RNAs, mostly spliceosomes. Around 15% of upregulated DEGs in SFN vs. CTR (i.e., 65 genes) are known to be associated with and/or are targets of NRF2. The same analysis in downregulated DEGs in BRU vs. CTR and upregulated DEGs in SFN vs. BRU revealed only 10% and 9% of the DEGs, respectively, known to be associated with NRF2. There were 64 DEGs that were contemporaneously upregulated in SFN vs. CTR and downregulated in BRU vs. CTR (Figure 2). Almost 23% (14 genes) of those DEGs are known to be associated with NRF2 (Appendix A). The gene with the largest difference in a positive response to SFN and a negative response to BRU was a non-coding RNA (LOC100848941), but among protein-coding genes *SLC7A11*, *ITPRIP*, *TFPI*, *HYAL1*, and *SERPINH1* were the ones with the highest difference in response to SFN and BRU.

Evaluating the expression of putative NRF2 target genes [17,18] in the RNAseq dataset and RTqPCR, there was a high similarity between these two techniques, albeit the expression of some genes had a different statistical significance (Figure 3).

In both techniques, SFN significantly increased the transcription of *GSR*, *KEAP1*, *NFE2L2*, *NQO1*, and *GCLC*; however, a significant decrease in the transcription of *GSTM1* by SFN treatment was captured only by RTqPCR. Both techniques disclosed an upregulation of *NFE2L2* and *GCLC* by BRU; however, significant decreases in the transcription of *GSR* and *KEAP1* by BRU treatment were only detected in RNASeq data, with both genes deemed to have high relevance as NRF2 targets by GeneCard (Appendix A).

Among all the NRF2 targets tested, only *NQO1, GSR,* and *KEAP1* had a pattern expected based on the treatments (i.e., downregulated by BRU and upregulated by SFN, compared to CTR); however, this was significant only in the RTqPCR for *NQO1* and RNAseq for *GSR* and *KEAP1.*

### 2.3. Functional Analysis of Differentially Expressed Genes

#### 2.3.1. SFN vs. CTR

The functional clustering and charts from the Database for Annotation, Visualization, and Integrated Discovery (**DAVID**) (Appendix A) revealed that cell cycle-related terms were the most enriched in downregulated DEGs in SFN vs. CTR, while among upregulated enriched DEGs were terms related to oxidative stress response with the NRF2 pathway being the most enriched term. Summary analyses of the Gene Ontology terms by REVIGO (Appendix A) further reiterated the overall enrichment of terms related to the cell response to oxidative stress followed by glycolytic processes in upregulated DEGs and cell division in downregulated DEGs.

Among the most enriched pathways (Figure 4) were the ones related to signaling and metabolism (especially glucose and amino acid) among upregulated DEGs and cell growth and death among downregulated DEGs. Other enriched pathways among the upregulated DEGs included ferroptosis, IL-17 signaling, and HIF-1 signaling. Several pathways associated with cellular survival, cell cycle, and DNA repair KEGG pathways were enriched in downregulated DEGs in MACT cells treated with SFN compared to the control. Downregulated DEGs associated with cell cycle, cellular senescence, and the p53 signaling pathway included *SMAD2*, *CCNA2*, *CCNB3*, *CCNB2*, and *CDK1*. The enrichment of pathways associated with DNA repair, including homologous recombination, DNA replication, mismatch repair, and base excision repair, was in part driven by the decreased expression of *POLD1*, the transcript encoding the catalytic subunit of the DNA polymerase enzyme, as well as *RFC2*, a transcript encoding replication factor subunits that assists in DNA elongation and DNA polymerase activity.

The analysis with the Dynamic Impact Approach (**DIA**, Figure 5, and Appendix A) confirmed induction of metabolism with other amino acids, with the most induced pathways being ‘Selenoamino acid metabolism’ and ‘glutathione metabolism’ and the inhibition of cell growth and death, particularly the ‘Cell cycle’ pathway, with induction of the ‘P53 pathway’. The analysis of upstream transcription factors (**UPTF**) revealed a large impact on the induction of the ETS transcription factor ERG [19], ZNF148, FOSL1, MAFG, MAFK, ATF4, and NFKB1, while FOXO3, MYBL2, and CUTL1 were inhibited by SFN treatment.

Super-PRED predicted 114 targets with the endoplasmic reticulum-associated amyloid beta-peptide-binding protein (**ERAB**), NRF2, NFKB1, and cathepsin D as the preeminent targets of SFN (Appendix A). The analysis of Enrichr revealed as the top pathways the transcriptional regulation NRF2 followed by the HSF1 (heat shock factor 1) among the upregulated DEGs in SFN vs. CTR (Appendix A). There were more than 300 transcription factors that were significantly (FDR < 0.05) enriched, with ATF3 being the most enriched.

#### 2.3.2. BRU vs. CTR

The functional clustering and charts from DAVID (Appendix A) revealed that upregulated DEGs with BRU vs. CTR were significantly associated with the regulation of transcription (both positive and negative regulation) and the nucleus while the downregulated DEGs were associated with cell adhesion and metabolism, particularly glucose metabolism, and organelles, such as the endoplasmic reticulum and Golgi, but they were also associated with extracellular interaction and the extracellular matrix. The analysis of KEGG pathways in upregulated DEGs revealed an enrichment of cell growth-, inflammation- (such as TNF signaling, apoptosis, and NF-kappa B signaling), and transcription-related pathways. Enriched pathways associated with downregulated DEGs in BRU vs. CTR were primarily associated with various metabolic functions, including various pathways associated with glycan biosynthesis, carbon metabolism, and amino acid metabolism. Both the citrate cycle as well as the pyruvate metabolism pathway were also enriched. Several signaling pathways were enriched in downregulated DEGs in BRU vs. CTR, including the ‘ECM–receptor interaction’, ‘mTOR signaling’, and ‘VEGF signaling’.

The DIA analysis (Figure 5 and Appendix A) revealed the strong impact and inhibition of BRU on metabolism, particularly for pathways related to energy metabolism (e.g., ‘oxidative phosphorylation’ and ‘nitrogen metabolism’), glycan synthesis (especially glycosaminoglycans), amino acid metabolism (especially related to Ala, Asp, Ile, Gly, Leu, Ser, Thr, and Val), lipid metabolism, and the metabolism of cofactors, with ‘vitamin B6 metabolism’ being the most impacted and inhibited pathway. Pathways that were also highly impacted related to the ‘Organismal system’, with the particular activation of pathways related to the immune system, particularly inflammatory-related pathways, such as the ones involving the NOD-like receptor, Toll-like receptor, RIG-I-like receptor, and T- and B-receptors. The analysis also revealed an activation of transcription and translation but also an inhibition of pathways involved with the folding, sorting, and degradation of proteins in the endoplasmic reticulum and cell cycle. Analysis of UPTF by DIA revealed that ATF4, ATF3, KLF6, and several transcription factors related to lipid metabolism, such as RXRB, SREBF1, and PPARG, were strongly inhibited, while NFATC1, FBXL10, SNAI2, and FOSL1 were strongly activated by BRU treatment.

Super-PRED (Appendix A) revealed that NRF2 is a demonstrated target of BRU and predicted 131 additional targets with MAP kinase ERK2, DNA-(apurinic or apyrimidinic site) lyase, the Bloom syndrome protein, hypoxia-inducible factor 1 alpha (HIF-1α), and NFKB as the most prominent. SFN and BRU shared 51 targets according to Super-PRED, with NRF2, NFKB, and cathepsin D as the ones with the largest overall probability.

Enrichr revealed that among the downregulated DEGs by BRU, NRF2 was not the most enriched UPTF; instead, the most enriched UPTF were RUNX2 and EGR1 (Appendix A). Similarly, the tool uncovered the most enriched pathways as the ones related to metabolism and the protein and immune system, but the NRF2 pathway was not significantly enriched with an FDR = 0.14.

#### 2.3.3. SFN vs. BRU

The functional clustering and charts from DAVID (Appendix A) revealed that upregulated DEGs in SFN vs. BRU were significantly associated with terms related to metabolic pathways, particularly in the endoplasmic reticulum, lysosome, and Golgi and were also related to carbohydrates and proteins with enrichment of redox homeostasis and response to oxidative stress. The downregulated DEGs in SFN vs. BRU were enriched with terms related to transcription and translation, cell cycle, and inflammation (e.g., NF-kappa B signaling) with genes related to the nucleus that interacted with RNA. The upregulated DEGs in SFN vs. BRU significantly enriched pathways associated with signaling, particularly mTOR and HIF-1 signaling, and metabolism, particularly related to glycosaminoglycans, amino acids, and glucose, while downregulated DEGs enriched pathways were associated with inflammation (e.g., TNF and NF-kappa B), cell growth and death, and transcription/translation (Figure 4).

The DIA analysis (Figure 5 and Appendix A) confirmed the above findings, revealing a strong activation of carbohydrates and amino acid metabolism-related pathways, the inhibition of transcription and translation, the cell cycle, the immune system, and particular pathways related to inflammation. The UPTF most impacted and induced was KLF6, followed by RXRB, SREBF1, ZEB1, and ATF4, while the most inhibited were SNAI2 and FOXO3.

The Enrichr tool uncovered NRF2 as the most enriched UPTF and KEAP1-NFE2L2 among the most enriched pathways (with HSF1 activation being the most enriched), with DEGs upregulated by SFN and downregulated by BRU (Appendix A).

## 3. Discussion

### 3.1. Gene Targets of NRF2 in MACT Cells

The findings of RT-qPCR and RNA-sequencing revealed that *NQO1, KEAP1,* and *GSR* are NRF2-specific targets in MACT cells. The expression of *NQO1* was strongly upregulated when cells were treated with SFN, as has previously been observed in other experiments [20,21], and downregulated when cells were treated with BRU (although only in RTqPCR). Prior research supports the important role of NRF2 in regulating *GSR* expression, as well as the presence of an ARE in the promoter region of the *GSR* gene [22], and KEAP1 is a known target of NRF2. While the general trends in the transcriptional profiles of putative NRF2 target genes were similar between the RTqPCR and RNAseq results, there were discrepancies between the two techniques in terms of which comparisons were found to be significant. The differences in significance may be driven by the limited number of samples, three, used in each experiment, and additional samples would likely bring the results from each technique into closer alignment. Another factor, particularly as it relates to the contrast in results regarding *GSTM1*, may be the overall low expression of the transcript. The average number of mapped reads across all samples for *GSTM1* was eight, with *GSTM1* being undetected in two of the samples. As discussed by Everaert et al. (2017), genes with low expression are more likely to yield different statistical results between RTqPCR and RNASeq investigations [23]. Furthermore, the normalization approach is very different between the two techniques, which can yield different results when the difference between groups is not large.

In our prior experiment, we determined that SFN and BRU are specific modulators of NRF2 [12]. However, these compounds do not only modulate NRF2, as BRU is known to inhibit the overall translational capacity in cells [24,25], as we confirmed in our prior study [12]. Thus, it is not a surprise that several transcription factors besides NRF2 were affected, as also indicated by the prediction of targets via Super-PRED.

Super-PRED predicted the ERAB pathway as the top target for SFN. This pathway is associated with the control of ER stress [26]. Our functional analysis uncovered the importance of the ER in both SFN (among upregulated DEGs) and BRU (among downregulated DEGs) treatments. NRF2 has a known role in controlling ER stress [27] and the ER stress increase in the mammary gland during lactation [28,29]; thus, our data suggest the possible positive role of NRF2 activation in reducing ER stress in mammary tissue.

The DIA analysis revealed a large number of UPTFs that were highly impacted by the SFN and BRU treatments. Among them, the ATF4 appeared to be prominent, as it was highly activated by SFN and highly inhibited by BRU. There is a known direct relationship between ATF4 and NRF2, as the former activates the latter in response to the integrated stress response [30].

Complicating the interpretation of the results is the known crosstalk of NRF2 with other transcription factors, including the hydrocarbon receptor, the retinoic X receptor alpha (RXRα), and NFkB [31]. Interestingly, the bioinformatic tools used revealed NFκB to be a prominent target for both SFN and BRU, which is a key transcription factor in the control of inflammation, among other functions [32]. Thus, the effect observed on inflammation might not be the consequence of a direct modulation of NRF2 only but a secondary effect by the crosstalk with NFκB, as one pathway can influence the activity of the other pathway [33]. The combined targeting of NRF2 and NFκB is actively pursued as a potential therapeutic approach for treating cancer [34]. This crosstalk modulates the transcription of target genes for these transcriptional factors.

Considering the above confounding effects of multiple UPTFs, the use of genes upregulated by SFN and contemporarily downregulated by BRU can aid in determining NRF2 target genes by looking into DEGs that are most affected by SFN (upregulated) and BRU (downregulated). Our analysis allowed us to identify 64 “novel” target genes. The soundness of such an approach was confirmed by the large number of known NRF2 targets and the analysis with the Enrichr tool, indicating NRF2 as the most enriched UPTF and pathway in this list of genes. The *SLC7A11*, coding for a cystine/glutamate antiporter, was the most affected. This gene is a well-known NRF2 target gene that works by mediating the ferroptosis inhibition that induces NRF2 [35]. Interestingly, the effect of ATF4 on NRF2 is also mediated by a cystine/glutamate antiporter (i.e., *SLC27A11*) [30]. The role of the protein coded by *SLC7A11* is not clear in mammary tissue; however, it has been implicated in the maintenance of intracellular glutathione levels and chemotherapeutic resistance in both ovarian [36] and lung cells [37]. Furthermore, an increase in the expression of amino acid transporters can benefit milk protein synthesis [38], although this amino acid transporter was the only one upregulated by SFN (see Appendix A). More than 75% of affected DEGs upregulated by SFN and downregulated by BRU, including the most affected (i.e., *ITPRIP*, *TFPI*, *HYAL1*, and *SERPINH1*), are not known NRF2 targets; thus, this can be considered novel.

The very high expression of some genes by SFN treatment is of interest. For instance, the very large increase in expression of *IL36A*, although coding for an inflammatory-related cytokine, is known to be an NRF2 target [39]. The *PHLDA2*, the second most upregulated DEGs by SFN, has no known role in NRF2; however, the *PHLDA1* that was upregulated by BRU (Appendix A) was determined to be an activator of NRF2 [40].

Together, these data provide novel insights into the NRF2 regulatory network in bovine mammary cells via the modulation by SFN and BRU. However, confounding effects on other transcription factors highlight limitations in the utilization of these modulators for the specific and direct targeting of NRF2 for nutrigenomic applications.

### 3.2. NRF2 Modulation Has Varying Effects on Glutathione System Genes

Most of the putative target genes evaluated in the RT-qPCR are associated with the glutathione system, and their expression levels varied from what was expected. Previous studies revealed NRF2 activity as essential for the upregulation of *GSTM1* in mice [41,42]. However, no effect was observed in MACT cells treated with SFN. Notably, *GSTP1*, the gene encoding for glutathione-S-transferase pi 1, was significantly upregulated when cells were treated with SFN compared to the control, with a tendency (*p* = 0.1) for lower *GSTP1* expression when cells were treated with BRU compared to the control. This may indicate species-specific differences in the NRF2-mediated regulation of specific glutathione-S-transferase isoforms.

The upregulation of *GCLC* transcription in response to SFN used in our study is consistent with previous work [42]; however, treatment with BRU also resulted in the increased expression of this gene. Given the presence of many different transcriptional regulatory elements in the promoter region of the *GCLC* gene [43], it is likely that another transcription factor, potentially *NFkB*, which was upregulated when cells were treated with BRU or *AP-1*, was responsible for this increase in the expression of *GCLC* when NRF2 was inhibited.

Although prior investigations have observed increases in *GPX1* expression in response to NRF2 activation [44,45], we observed no increase in *GPX1* expression when cells were treated with sulforaphane. Additional studies on *GPX1* expression have found that other transcription factors, such as TFAP2C [46] and RORα [47], can regulate *GPX1* expression, suggesting that NRF2-dependent regulation may only occur in conditions of heightened oxidative stress. Similar to what was found regarding *GSTP1*, *GPX2* was the only glutathione peroxidase gene that was upregulated when cells were treated with SFN compared to the control; however, none of the glutathione peroxidase genes were affected by their treatment with BRU. Again, this may suggest that the NRF2-mediated transcriptional regulation of particular glutathione system genes happens in a species-specific manner.

The lack of consistency between the results from our study and previous studies, particularly those regarding the SFN activation of NRF2, may be related to the enrichment of the p53 signaling activity detected in the functional analysis with genes downregulated by SFN treatment compared to the control. Notably, *CCNB1*, the gene encoding for cyclin B1, is associated with the p53 signaling pathway and was downregulated when cells were treated with SFN. Prior studies have demonstrated that silencing *CCNB1* results in the increased expression of p53 [48], and Faraonio et al. (2006) found that p53 activity can interfere with the ability of NRF2 to bind ARE and activate the transcription of these genes [49]. This may serve as an explanation for why genes like *GSTM1* and *GPX1* were unaffected when treated with sulforaphane.

### 3.3. Cellular Stress and Feedback Loops Regulate NRF2 and KEAP1 Expression

Based on the findings of the functional analysis, it is clear that both of the treatments generated some degree of cellular stress, with BRU having the greatest effect. The increase in the expression of NRF2 (gene name *NFE2L2*) observed by both BRU and SFN treatments likely occurred in two different ways. First, NRF2 is known to contain ARE sequences in its promoter sequence, and prior studies have shown that NRF2 can activate its own transcription, generating a positive feedback loop [50]; on the other hand, BRU was found to have a strong activating effect on genes associated with the NF-kB signaling pathway, and it is known that NF-kB can activate the transcription of NRF2 [51]. To prevent the positive feedback loop of NRF2 from promoting its own transcription and from leading to unregulated signaling, *KEAP1*, the negative regulator of NRF2 activity, is also transcribed under conditions of NRF2 activation [52], which is consistent with what we found in the RT-qPCR and RNA sequencing analysis where *NFE2L2* and *KEAP1* transcription were increased when cells were treated with SFN. Notably, *KEAP1* transcription was determined to be upregulated by SFN by both RT-qPCR and RNAseq but was found to be significantly downregulated in RNAseq only when cells were treated with BRU. This may indicate that the potential activation of NRF2 expression by BRU via alternative pathways like NFkB may not necessitate the upregulation of *KEAP1*, as seen with SFN.

### 3.4. SFN Treatment Promotes NRF2-Related Pathways While Inhibiting the Cell Cycle and Affecting mRNA Splicing

In support of our hypothesis, the treatment of immortalized bovine mammary cells with SFN resulted in the increased expression of genes associated with the canonical roles of NRF2, including response to oxidative stress. The dual role of NRF2 in both preventing tumorigenesis by preventing the deleterious effects of ROS as well as promoting tumor survival by enhancing tumor survival in hypoxic conditions [53] is an important topic in cancer research. Our findings support these roles of NRF2, as the treatment of cells with sulforaphane resulted in the downregulation of genes associated with cell cycle progression and DNA repair. The increased expression of genes associated with proteasome function is likely a consequence of the negative feedback loop responsible for the regulation of the NRF2 response [52], as the proteasomal degradation of NRF2 is an important mechanism regulating its activity.

It is also of interest that among the most upregulated transcripts were the ones related to spliceosomes. NRF2 has been demonstrated to play an important role in controlling alternative splicing [54]. Alternative splicing is recognized to have an important role in controlling the transcriptome and, as a consequence, cellular functions and can also have epigenetic effects [55,56]. Thus, the change in the expression of genes by the activation of NRF2 by SFN may also be partly regulated by the effect of alternative splicing. The role of alternative splicing in the regulation of milk synthesis in the mammary gland was revealed by recent studies [57,58,59].

### 3.5. The General Protein Synthesis Inhibitor Brusatol Negatively Affects Metabolism While Stimulating Inflammatory Pathways

Given the role of BRU as a general protein synthesis inhibitor [24], it was not surprising that it affected the greatest number of genes, with a strong effect on pathways associated with metabolism and inflammation. However, surprisingly, the bioinformatic analyses (for both DAVID and DIA) of the transcriptomic data suggested an increase in protein translation with BRU instead of inhibition, although a decrease in protein synthesis might be indicated by the downregulation of mTOR pathway [60]. A decrease in protein synthesis capacity when cells were treated with BRU was also supported by our prior observation of lower luciferase and renilla signaling [12]. Under conditions in which protein translation was inhibited, the metabolism and biosynthesis of glycans and amino acids, as well as the TCA cycle, were inhibited. Furthermore, the downregulation of genes associated with protein processing in the endoplasmic reticulum may have made the cells susceptible to protein misfolding or the activation of the unfolded protein response. The relationship between the unfolded protein response and inflammation [61] was potentially a cause for the increase in the expression of genes associated with the NF-kB and TNF signaling pathways when cells were treated with BRU.

## 4. Materials and Methods

### 4.1. Reagents and Cell Culture

Sulforaphane (SFN) (Cat #S4441, Sigma-Aldrich, St. Louis, MO, USA) and Brusatol (BRU) (Cat #SML1868, Sigma-Aldrich) were solubilized in DMSO and used to treat cells based on our prior findings [12]. All treatments were prepared in traditional cell culture growth media composed of high-glucose Dulbecco’s Modified Eagle’s Medium (DMEM) with sodium pyruvate (Cat #25-500, Genesee Scientific, San Diego, CA, USA) and 10% fetal bovine serum (Cat #F4135, Genesee Scientific). Before treatment, MACT cells were seeded in 6-well plates at a seeding density of 3 × 10^5^ cells per well in DMEM media and grown overnight in an incubator at 37℃/5%CO_2_. Three replicates of MACT cells were then treated with either 10 µM SFN, 100 nM BRU, or an equivalent amount of DMSO (i.e., control) for 24 h before the isolation of RNA. Cells were cultured, and treatments were applied in 2 mL of media/well.

### 4.2. RNA Isolation and RTqPCR of Putative Target Genes

RNA was isolated using the RNeasy Kit (Cat#74004, QIAGEN, Redwood City, CA, USA). The synthesis of complementary DNA (cDNA), RTqPCR, and data analysis by LinRegPCR were all performed in accordance with a previously published protocol [62]. Primers for glutathione reductase (*GSR*), glutathione S-transferase M1 (*GSTM1*), Kelch-like ECH-associated protein 1 (*KEAP1*), nuclear factor, erythroid 2-like 2 (*NFE2L2*), glutamate-cysteine ligase catalytic subunit (*GCLC*), glutathione peroxidase 1 (*GPX1*), and NAD(P)H quinone dehydrogenase 1 (*NOQ1*) were taken from a previous bovine study [17]. Four reference genes, Glyceraldehyde 3-phosphate dehydrogenase (*GAPDH*), Ribosomal Protein S9 (*RPS9*), Mitochondrial Ribosomal Protein L39 (*MRPL39*), and the ubiquitously expressed transcript (*UXT*), were tested using GeNorm [63]. The normalization factor was obtained using the geometrical mean of *RPS9*, *MRPL39,* and *UXT* with a V-value of 0.12. *GAPDH* was deemed to be unreliable as a reference gene, and thus, it was used as a target gene.

### 4.3. RNA Sequencing and Analysis

RNA isolated from the treated cells was checked for integrity using Agilent TapeStation 4200. The RIN was >8.1 (9.1 ± 0.5) for all samples. RNA was sent to the Center for Genome Research and Bioinformatics at Oregon State University for high-throughput sequencing. Library construction was performed utilizing a QuantSeq 3′ mRNA-Seq Library Prep Kit FWD for Illumina (015.96, Lexogen, Greenland, NH, USA). Sequencing was performed on an Illumina HiSeq3000 platform at 60 samples/lane. Quality control was performed, as previously described [64]. Differential gene expression analysis was performed using the DESeq2 [65] package in RStudio (v1.4). Differentially expressed genes were denoted as genes with a false discovery rate (FDR) < 0.05. Raw data were submitted to the Sequence Read Archive from NCBI, submission #SUB14257804.

### 4.4. Bioinformatics Analysis

The enrichment of GO terms and pathways by DEGs was performed using the DAVID online functional annotation tool [66] and the Dynamic Impact Approach [67].

For DIA, the analyses of KEGG pathways and the upstream transcription factor [68] were performed. The functional analysis in DAVID was performed with all DEGs and up- and downregulated DEGs in each comparison. The analysis was run using the default setting, and an EASE score of ≤0.05 was used as a cut-off to determine significantly enriched terms. The default setting plus the WIKIPATHWAYS were used for the analysis, and results were downloaded as the functional annotation clustering and chart. Gene Ontology terms from the results of DAVID were summarized and visualized using REVIGO with the results listed as small (0.5) and using the *p*-value of the enrichment of GO terms and the dataset for *Bos taurus* [69]. The ChEA 2022 transcription and the pathways of Enrichr [70] were also used to analyze upregulated DEGs by SFN, downregulated DEGs by BRU, and upregulated DEGs in SFN vs. BRU in the attempt to characterize potential NRF2 target genes. The whole detected transcriptome was used as the reference or background database for all bioinformatic analyses.

The Venn diagram was made using a free available tool (https://bioinformatics.psb.ugent.be/webtools/Venn/, accessed on 5 March 2024). NRF2 target genes were analyzed by the WEB-based Gene Set Analysis Toolkit (https://www.webgestalt.org/, accessed on 5 March 2024), and GeneCards were analyzed by searching for known NRF2 targets. The target prediction of the anatomical therapeutic chemical for SFN and BRU was run using Super-PRED (https://prediction.charite.de/index.php, accessed on 9 March 2024).

### 4.5. Statistical Analysis

The statistical analysis of RT-qPCR data was performed using the PROC GLM procedure in SAS v9.4 (SAS, Cary, NC, USA), with treatment as the main effect. All data were log_2_-transformed prior to statistical analysis.

## 5. Conclusions

Overall, the findings of this investigation indicate that activation of NRF2 with SFN results in the upregulation of many putative NRF2 target genes and associated pathways, with additional effects on metabolic and cell cycle pathways. Inhibition with the non-specific NRF2 inhibitor BRU affected a wide range of genes and pathways, notably activating pathways associated with inflammation and cellular stress while inhibiting various metabolic pathways. Taken together, these results highlight potential species-specific differences in the NRF2 regulatory network between bovine and other species, as well as demonstrate potential limitations with regard to nutrigenomic therapies for increasing NRF2 activity in dairy cows. Future work assessing the efficacy of natural NRF2 agonists and antagonists in in vivo investigations is critical for understanding the implications of NRF2 modulation within dairy cows as well as for understanding the additional effects that natural NRF2 modulators have at the systemic level.

## Figures and Tables

**Figure 1 ijms-25-04264-f001:**
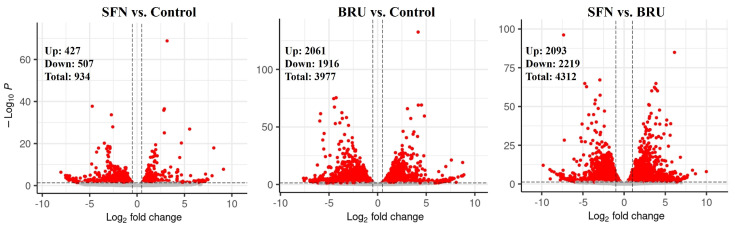
Treatment of MACT cells with BRU has a larger effect on the transcriptome than treatment with SFN. Volcano plots of differentially expressed genes in each comparison (NRF2 activator sulforaphane = SFN; NRF2 inhibitor brusatol = BRU). Dots in red were considered differentially expressed genes on the basis of having a log2 fold change > |0.5| and FDR < 0.05.

**Figure 2 ijms-25-04264-f002:**
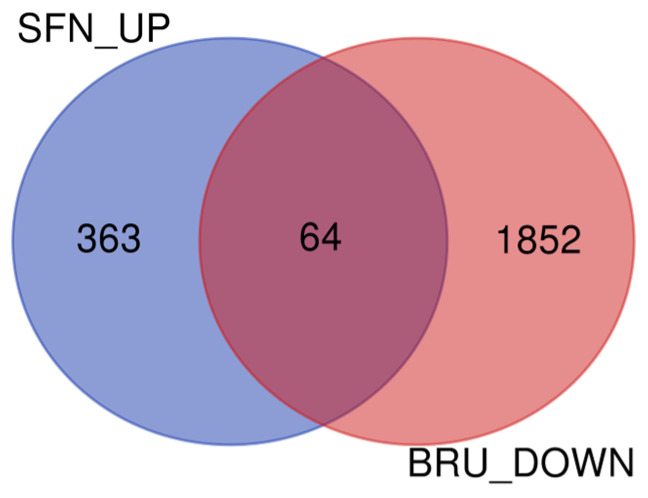
Venn diagram of the differentially expressed genes (FDR < 0.05) that were upregulated in SFN vs. CTR and downregulated in BRU vs. CTR.

**Figure 3 ijms-25-04264-f003:**
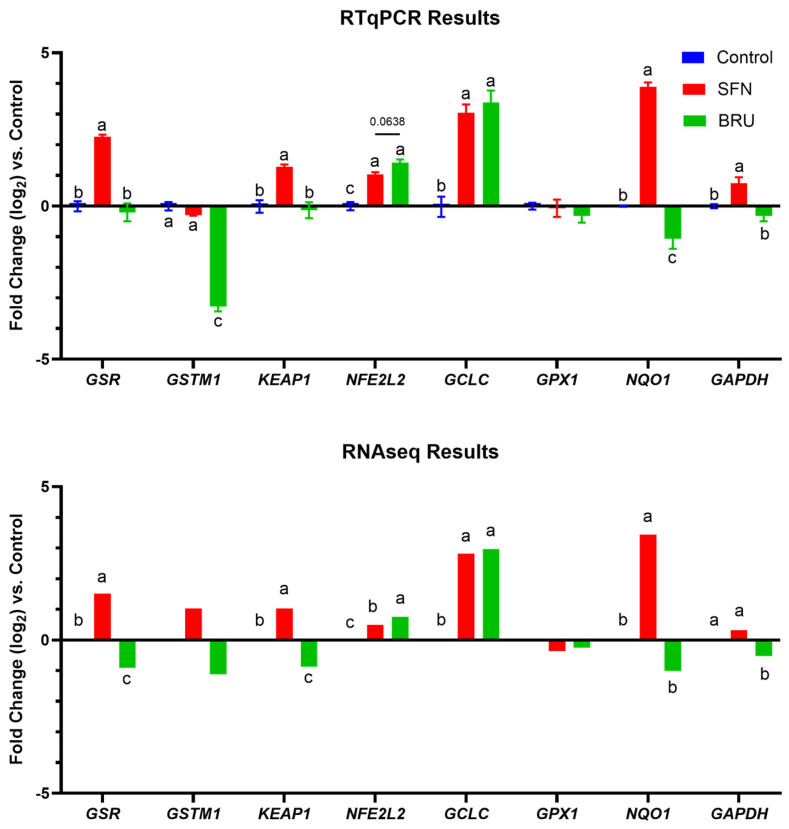
General trends in gene expression are similar between RTqPCR and RNASeq analyses, with statistical differences in certain genes. The expression of putative NRF2 target genes, as determined by RT-qPCR and RNAseq. Expression values are displayed as log2 fold change compared to the control treatment. Letters (a, b, c) denote significant differences for each transcript. Significance was declared at *p* < 0.05, with *p*-value indicated for tendencies.

**Figure 4 ijms-25-04264-f004:**
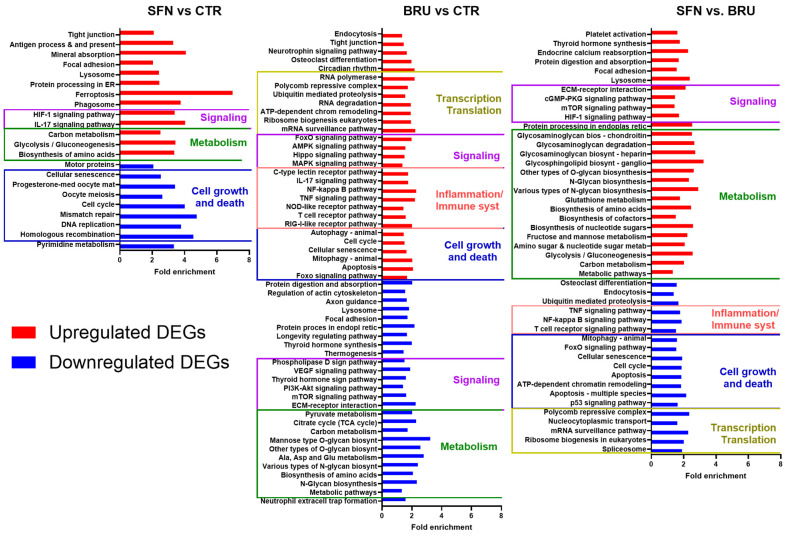
DAVID analysis highlights the functional implications of SFN and BRU treatment on the transcriptome of MACT cells. Significantly (EASE score < 0.05) enriched KEGG pathways in DAVID associated with up-(red) and down-(blue) regulated DEGs in MACT cells treated with sulforaphane (SFN) or brusatol (BRU) compared to non-treated cells (CTR), and the DEGs between SFN- and BRU-treated cells. The X-axis provides fold enrichment using the DAVID bioinformatic tool.

**Figure 5 ijms-25-04264-f005:**
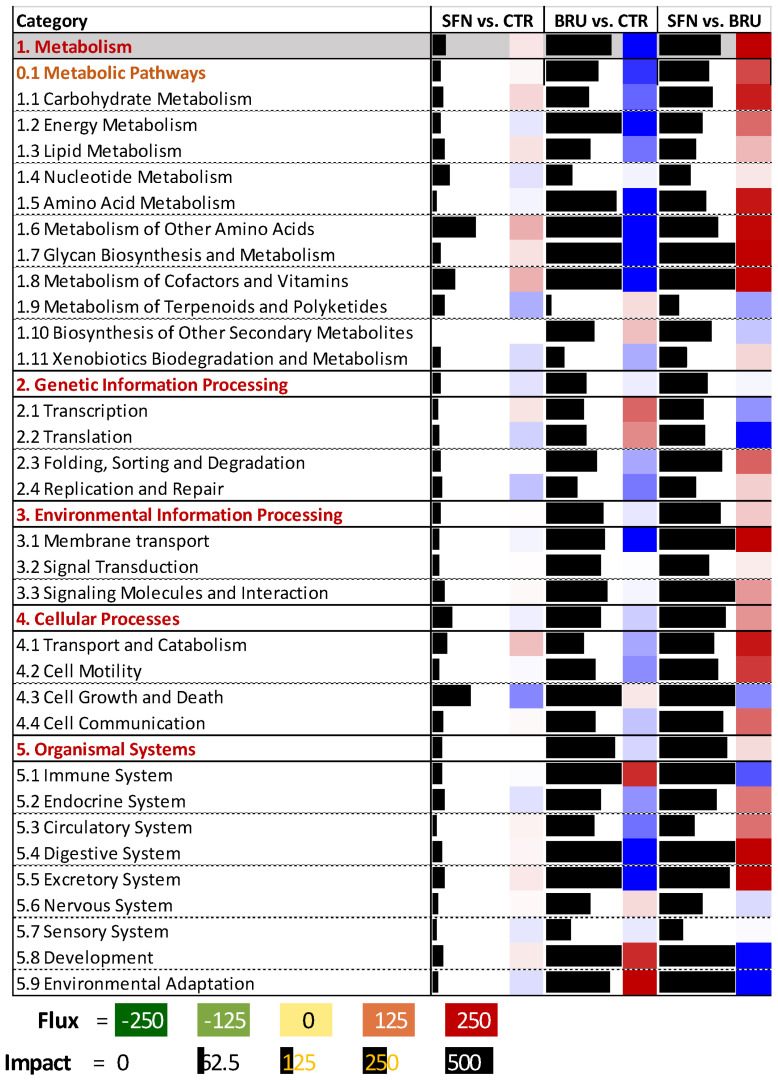
DIA analysis identifies the effects of SFN and BRU treatments on the direction and magnitude of functional changes in MACT cells. Summary of categories of KEGG pathways as analyzed by the Dynamic Impact Approach. Shown are the impact (black horizontal bar; the larger the bar, the larger the impact) and the flux (or direction of the impact; red denotes activation and blue inhibition) of the category of pathways for MACT cells treated with sulforaphane (SFN) or brusatol (BRU) compared to non-treated cells (CTR) and between SFN- and BRU-treated cells.

## Data Availability

The raw RNAseq data presented in this study are openly available at the NCBI Sequence Read Archive; data were submitted (submission # SUB14257804) awaiting approval. The raw RTqPCR data supporting the conclusions of this article will be made available by the authors on request.

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
