# Peer review of "The Experimental and In Silico-Based Evaluation of NRF2 Modulators, Sulforaphane and Brusatol, on the Transcriptome of Immortalized Bovine Mammary Alveolar Cells"

_ijms, 2024, doi:10.3390/ijms25084264_

Round 1

Reviewer 1 Report

Comments and Suggestions for Authors

Nice and interesting work. Thanks for their focus. But should be more focused.

Following issues should be revised and corrected:

The title should be as: The experimental and in silico-based evaluation of NRF2 modulators sulforaphane and 2 brusatol on the transcriptome of bovine mammary gland cell line.

One of the big issues in this study is the limitation of such broad inadequate experimental work. Please elaborate this issue in the discussion part. Another problem is the lack of strong conclusion! I would also add a schematic mechanistic figure such study in the text.

Conclusion both in the abstract and other parts of the MS should be straightforwardly appeared.

The design part is poorly graspable.

Lines 55-53, ….mainly by the nuclear factor erythroid 2 related factor 2 (NRF2)[8].

Line 73, …. “ruminants”…. For …..”ruminant animals”; also throughout the text (line 74 etc..).

Lines 146-147, How the … “Other enriched pathways include ferroptosis, IL-17 signaling, and HIF-1 signaling” I related [specially the IL-17)?

Line 175, … the know crosstalk of NRF2 with…?

Author Response

The title should be as: The experimental and in silico-based evaluation of NRF2 modulators sulforaphane and 2 brusatol on the transcriptome of bovine mammary gland cell line.

Thank you for the suggestion – the title has been changed to “The experimental and in silico-based evaluation of NRF2 modulators sulforaphane and brusatol on the transcriptome of immortalized bovine mammary cells”

One of the big issues in this study is the limitation of such broad inadequate experimental work. Please elaborate this issue in the discussion part. Another problem is the lack of strong conclusion! I would also add a schematic mechanistic figure such study in the text.

A brief discussion of this topic has been added to the beginning of the discussion and a graphical abstract has been added. WE hope with this we can clarify the experimental design.

Conclusion both in the abstract and other parts of the MS should be straightforwardly appeared.

Thank you for the revision. The conclusion has been more clearly outlined in the abstract (line 25). A brief concluding paragraph regarding section 3.1 has been added into the discussion on lines 319-322. The conclusion has been moved to appear after the discussion in the manuscript and has been updated (lines 411-423)

The design part is poorly graspable.

We hope that with the graphical abstract, the design is clear

Lines 55-53, ….mainly by the nuclear factor erythroid 2 related factor 2 (NRF2)[8].

Thank you for the revision; the text has been updated

Line 73, …. “ruminants”…. For …..”ruminant animals”; also throughout the text (line 74 etc..).

Thank you for pointing this out. Changes have been made throughout the manuscript

Lines 146-147, How the … “Other enriched pathways include ferroptosis, IL-17 signaling, and HIF-1 signaling” I related [specially the IL-17)?

Additional context has been added to this statement indicating that these pathways are associated with the upregulated genes in the SFN vs. CTR treatment.

Line 175, … the know crosstalk of NRF2 with…?

We could not identify this text near line 175, but found the word “know” and  replaced it with “known” . Thank you for indicating this revision.

Reviewer 2 Report

Comments and Suggestions for Authors

In this study authors evaluated of brusatol and sulforaphane (two modulators of NRF2 signaling) on the transcriptome of immortalized bovine mammary alveolar cells (MACT). Authors found that treatment with SFN resulted in the activation of many putative NRF2 target genes and the upregulation of genes associated with pathways involved in cell survival, metabolism, and antioxidant function while suppressing the expression of genes related to cellular senescence and DNA repair. Contrarely, treatment with brusatol resulted in the upregulation of genes associated with inflammation, cellular stress, and apoptosis while suppressing the transcription of genes involved in various metabolic processes.

the manuscript is interesting and generally well written. Figures are of good quality and easily readable. However, there are some important points to improve. In particular:

Line 89: Please define FDR

Lines 294-295: One of the most important role of  SLC27A11 is the production of glutathione (GSH), an important antioxidant (see PMID: 38203758)

Figure 3: How do the authors explain this important diffence in SFN and BRU regulation of NRF2/KEAP1 signaling and its target genes? In particular, how do the authors explain this different results between RTqPCR and RNAseq?

Figure 5: What is the sense of comparing SFN vs BRU if each treatment must be compared with the untreated control? 

2.3.3. SFN vs. BRU:  What is the sense of comparing SFN vs BRU if each treatment must be compared with the untreated control? 

4. Materials and Methods: Culture conditions of  MACT cells must be added. 

Line 402: Authors must state how many times the experiments have been repeated

Abbreviations must be written in full length when mentioned for the first time

Author Response

Line 89: Please define FDR

Thank you for point this out, FDR has been spelled out as false discovery rate

Lines 294-295: One of the most important role of  SLC27A11 is the production of glutathione (GSH), an important antioxidant (see PMID: 38203758)

Thank you for highlighting this reference. We have added information from this reference as well as another reference on SLC7A11 to the discussion.

Figure 3: How do the authors explain this important diffence in SFN and BRU regulation of NRF2/KEAP1 signaling and its target genes? In particular, how do the authors explain this different results between RTqPCR and RNAseq?

Thank you for acknowledging this. We have added a section to the discussion offering an explanation as to why the statistical results between the RNASeq and RTqPCR experiments were different.

Figure 5: What is the sense of comparing SFN vs BRU if each treatment must be compared with the untreated control?

Thank you for the comment. The purpose of comparing SFN v BRU treatments was to provide insight into the NRF2-regulatory network. By comparing SFN treatment, which activates NRF2, and BRU treatment, which inhibits NRF2, we aimed to identify genes and pathways that may be specifically regulated by changes in NRF2 activity, particularly by looking at genes that were up-regulated by SFN and down-regulated by BRU. We have provided relevant information on these genes/pathways in the first part of the discussion (3.1) while also highlighting the confounding effect that other transcription factors and regulatory networks may have.

2.3.3. SFN vs. BRU:  What is the sense of comparing SFN vs BRU if each treatment must be compared with the untreated control?

Thank you for your question. See the response to the previous question for why we believe it was important to consider this comparison.

  1. Materials and Methods: Culture conditions of MACT cells must be added.

Thank you for highlighting this revision. We have added additional information on the cell culture conditions in section 4.1.

Line 402: Authors must state how many times the experiments have been repeated

We have stated that three replicates of MACT cells were used in each treatment

Abbreviations must be written in full length when mentioned for the first time

We have addressed the point as indicated

Reviewer 3 Report

Comments and Suggestions for Authors

The authors have made a good effort to conduct, compile and draft the findings of the study entitled “Effect of the natural NRF2 modulators sulforaphane and 2 brusatol on the transcriptome of immortalized bovine mammary cells”. The RNAseq data revealed some interesting finding on how the NRF2 modulators influenced the downstream processing, activating and/or inhibiting varied pathways and mechanisms. Below mentioned are some minor suggestions

·      Line 121-124: In this sentence, the authors initially stated both the techniques (transcriptomics and RTqPCR) to “identify significant increase the transcription of GSR, KEAP1, NFE2L2, GCLC, and NQO1 due to SFN” while in the same sentence they mention a down-regulation (indirectly can be understood as decreased transcription) of GSTM1 and NQO1 in RTqPCR and likewise decreased transcription of GSR and KEAP1 in RNAseq data. This statement seems to be contradicting. Kindly confirm this.

·      Likewise, what was the expression profile of the genes uniquely expressed in either of the techniques (RTqPCR or RNAseq), for e.g. GSTM1 and NQO1

·      Line 162: “revealed a large impact an induction” was that a typing error? Did the authors mean, “revealed a large impact on induction

·      Line 339: the following citation was missed to be numbered- Faraonio et al., (2006)

Author Response

Line 121-124: In this sentence, the authors initially stated both the techniques (transcriptomics and RTqPCR) to “identify significant increase the transcription of GSR, KEAP1, NFE2L2, GCLC, and NQO1 due to SFN” while in the same sentence they mention a down-regulation (indirectly can be understood as decreased transcription) of GSTM1 and NQO1 in RTqPCR and likewise decreased transcription of GSR and KEAP1 in RNAseq data. This statement seems to be contradicting. Kindly confirm this.

Thank you for indicating this. This section has been clarified and better reflects the findings demonstrated in Figure 3.

Likewise, what was the expression profile of the genes uniquely expressed in either of the techniques (RTqPCR or RNAseq), for e.g. GSTM1 and NQO1

We have added into the discussion about possible reasons for the discrepancy between the two techniques, especially for low-expressed genes.

Line 162: “revealed a large impact an induction” was that a typing error? Did the authors mean, “revealed a large impact on induction”

Yes, thank you for pointing this out. The sentence has been changed.

Line 339: the following citation was missed to be numbered- Faraonio et al., (2006)

The reference is now numbered.

Round 2

Reviewer 1 Report

Comments and Suggestions for Authors

This nice and interesting work should finally be appeared in Int. J. Mol. Sci., of course after further revision for clarity, readability and correction.

The suggested more appropriate title should be as:

-“The experimental and in silico-based evaluation of NRF2 modulators sulforaphane and brusatol on the transcriptome of bovine mammary gland cell line” [OR] “The experimental and in silico-based evaluation of NRF2 modulators sulforaphane and brusatol on the transcriptome of immortalized bovine mammary gland epithelia”

-It would be more specified to mention bovine “mammary epithelial cells” for “mammary alveolar cells” specially some key parts throughout the text [of course after definition]; as such the readers would be led more appropriately correctly.

-The novelty of the study should be more highlighted [specially in the introduction part] (what it add for your previous nice work ref#17 and others)....?

- The vigor of figs and their related captions along with their clarity should be strengthened.

-Please use better and simpler general English terms through the text, such as used for utilized (line 43) etc.. also, many other words throughout; like “Prior to” ..... should be replaced by “before” ....etc.; please elaborate very precisely in the revision process.

-I would replace “ruminants”…. For ….. “ruminant animals” throughout the text.

- Lines 77-84, hard to grasp to readers. I would split this paragraph with two or more simpler sentence(s).

-About the captions of figs. specially 1, 3, 4 and 5; please mention what is the main message of each; readers might want to immediately grasp. Also some parts it is necessary to mention the n=mean±SD?SEM? How many? Etc..

Lines 253-254, it is introduction-based notion; please remove it from discussion part.

Lines 446-448, how much was the volume per well for your 6-well plates? Please specify? I would use 3ml/well?

Good luck

Author Response

Response to Reviewers:

We greatly appreciate the reviewers taking the time to continue their evaluation of our manuscript. We have provided responses to the revisions described by the reviewers below and have highlighted the associated revisions and changes in the updated submission of the manuscript.

Responses to Reviewer 1:

The suggested more appropriate title should be as:

“The experimental and in silico-based evaluation of NRF2 modulators sulforaphane and brusatol on the transcriptome of bovine mammary gland cell line” [OR] “The experimental and in silico-based evaluation of NRF2 modulators sulforaphane and brusatol on the transcriptome of immortalized bovine mammary gland epithelia”

Thank you for the suggested revision. We have added the word “alveolar” between mammary and cells in the title.

It would be more specified to mention bovine “mammary epithelial cells” for “mammary alveolar cells” specially some key parts throughout the text [of course after definition]; as such the readers would be led more appropriately correctly.

We appreciate the reviewer for pointing this out in the text. The MACT cell line used in this study is a mammary alveolar cell line, which is indeed an epithelial cell. MACT is the name given by the researchers who isolated the cells (see https://www.sciencedirect.com/science/article/pii/001448279190422Q). As such, we believe it is important to maintain the original name. We agree that it should be specified that those are epithelial cells, and we have now indicated that those are epithelial cells the first time that are mentioned in the main body of the manuscript.

The novelty of the study should be more highlighted [specially in the introduction part] (what it add for your previous nice work ref#17 and others)....?

This has been clarified on lines 73-77 to highlight better the novelty and importance of the research performed in this study.

The vigor of figs and their related captions along with their clarity should be strengthened.

We do not understand what the reviewer is highlighting here. The figures are all of good quality for publications. We have attempted to improve the caption of the figures (see below).

Please use better and simpler general English terms through the text, such as used for utilized (line 43) etc.. also, many other words throughout; like “Prior to” ..... should be replaced by “before” ....etc.; please elaborate very precisely in the revision process.

We have revised the text in the manuscript to increase readability.

I would replace “ruminants”…. For ….. “ruminant animals” throughout the text.

The word “ruminants” is no longer in the text and has been replaced with “ruminant animals”.

Lines 77-84, hard to grasp to readers. I would split this paragraph with two or more simpler sentence(s).

We have split this into two paragraphs to improve clarity and readability.

About the captions of figs. specially 1, 3, 4 and 5; please mention what is the main message of each; readers might want to immediately grasp. Also some parts it is necessary to mention the n=mean±SD?SEM? How many? Etc..

More descriptive title statements have been included in figures 1,3,4 and 5 to better describe the results that are depicted.

Lines 253-254, it is introduction-based notion; please remove it from discussion part.

We have removed this section from the discussion.

Lines 446-448, how much was the volume per well for your 6-well plates? Please specify? I would use 3ml/well?

We have specified on lines 443-444 that cells were cultured and treatments applied in 2mL/well.

Reviewer 2 Report

Comments and Suggestions for Authors

the manuscript has been significantly improved and can be accepted in the present form 

Author Response

We thank the reviewer for revisiting our manuscript.